# Open Pore Ultrafiltration Hollow Fiber Membrane Fabrication Method via Dual Pore Former with Dual Dope Solution Phase

**DOI:** 10.3390/membranes12111140

**Published:** 2022-11-13

**Authors:** Kyunghoon Jang, Thanh-Tin Nguyen, Eunsung Yi, Chang Seong Kim, Soo Wan Kim, In S. Kim

**Affiliations:** 1School of Earth Sciences and Environmental Engineering, Gwangju Institute of Science and Technology, 123 Cheomdangwagi-ro, Buk-gu, Gwangju 61005, Korea; 2Inosep Inc., E3 BLDG-408, 123 Cheomdangwagi-ro, Buk-gu, Gwangju 61005, Korea; 3Department of Internal Medicine, Chonnam National University Medical School, 160 Baekseo-ro, Dong-gu, Gwangju 61469, Korea

**Keywords:** hollow fiber, dual layer, dual pore former, open pores, finger-like structure

## Abstract

Hollow-fiber membranes are widely used in various fields of membrane processes because of their numerous properties, e.g., large surface area, high packing density, mass production with uniform quality, obvious end-of-life indicators, and so on. However, it is difficult to control the pores and internal properties of hollow-fiber membranes due to their inherent structure: a hollow inside surrounded by a wall membrane. Herein, we aimed to control pores and the internal structure of hollow-fiber membranes by fabricating a dual layer using a dual nozzle. Two different pore formers, polyethylene glycol (PEG) and polyvinyl pyrrolidone (PVP), were separately prepared in the dope solutions and used for spinning the dual layer. Our results show that nanoscale pores could be formed on the lumen side (26.8–33.2 nm), and the open pores continuously increased in size toward the shell side. Due to robust pore structure, our fabricated membrane exhibited a remarkable water permeability of 296.2 ± 5.7 L/m^2^·h·bar and an extremely low BSA loss rate of 0.06 ± 0.02%, i.e., a high BSA retention of 99.94%. In consideration of these properties, the studied membranes are well-suited for use in either water treatment or hemodialysis. Overall, our membranes could be considered for the latter application with a high urea clearance of 257.6 mL/min, which is comparable with commercial membranes.

## 1. Introduction

Membranes are essential parts of various modern industries, including water treatment [1], desalination [2], medical appliances [3], food and beverages [4], pharmacy [5], and energy fields [6,7]. This is because membranes prevent the movement, mixing, or delaying of two different phases and selectively transfer only specific substances. These membranes are classified in various ways according to their materials and structures. Among them, the porous membrane is composed of a solid phase with multitudinous pores. Porous membranes are distinguished by their ability to filter substances based on the size distribution of their pores. According to pore size, the membranes are classified as microfiltration (MF), ultra-filtration (UF), nano-filtration (NF), and reverse osmosis (RO) [8]. This means that pore size plays a crucial role in selecting a proper membrane for sufficiently separating desired contaminants. Therefore, various manufacturing methods such as phase-inversion [2], sintering [9], stretching [10], track-etching [11], template-leaching [12], interfacial reaction [13], sol-gel [14], and electro-spinning [15] have been developed and studied to control the pore size of porous membranes.

Membrane fabrication using phase inversion was first introduced by Loeb in 1962 [16]. The convenient and easy scaling of this method has made it popular in manufacturing polymer membranes. Phase-inversion methods include non-solvent-induced phase separation (NIPS), thermally induced phase separation (TIPS), vapor-induced phase separation (VIPS), evaporation-induced phase separation (EIPS), and the like. As the polymer in a solvent is exposed to an environment with low solubility, it will contact/mix with a non-solvent. Consequently, the polymer is precipitated from the solvent and a porous structure is created. Particularly, this process is easily used for the manufacture of a separation membrane with a specific shape, such as a hollow-fiber (HF) membrane [17,18,19].

Based on their configuration, membranes are classified as flat sheet, spiral wound, tubular, capillary, hollow fiber, etc. Among them, the HF membrane has a high effective area compared to its volume, leading to a high packing density. This is because the HF membranes possess a hollow in their middle and have a cylindrical long-strand structure [20,21,22]. HF membranes have been applied to various applications, such as water treatment filters, concentration filters, and hemodialysis filters, due to their outstanding characteristics. HF membranes are mainly manufactured using phase-inversion via the NIPS method. In NIPS, a polymer dissolved in a solvent is mixed with a non-solvent, and in the process of polymer precipitation, a polymer-rich phase and a polymer-lean phase govern the porous structure of the polymer membrane [23]. This manufacturing method can be easily applied for the mass production of a porous polymer membrane with an identical structure. However, it is difficult to produce an HF membrane with a uniform pore size; this is due to the change in diffusion rate that occurs during the demixing of the solvent. Therefore, in the NIPS process, adding additives to uniformly control the pores has been extensively explored. For example, several studies have explored the mixing of water-soluble polymeric pore former (WSP) in polymer solution, which is an expedient method for the selective decomposition of nanostructures, as a means of controlling structure via block copolymer; notably, WSP has been widely utilized in NIPS membrane fabrication, since this method may help to form uniform pores. Typically, this approach is governed by controlling the content of the pore former, and it has the advantage of good reproducibility and easy mass production [24,25].

Typical WSPs are polyethylene glycol (PEG), poly (vinyl pyrrolidone) (PVP), polyvinyl alcohol (PVA), etc. and help to promote solvent demixing at the interface between polymer solution and non-solvent to control the size of surface pores on a nanoscale. However, the pore-size control process using WSP has a difficulty in the formation of uniform pores due to the contact interface instability caused by the change in the concentration of the non-solvent and the fluidity of the solution. In addition, as most of the WSP diffuses and is removed in the direction of the contact interface, polymer-rich and polymer-lean phases are irregularly formed on the substrate. This causes an irregular structure of HF decreasing the mass transfer of the solute [24,26,27]. 

Herein, our work also applied the NIPS process with WSP for HF membrane fabrication. However, to overcome the aforementioned limitations, a dual-layer HF membrane was fabricated by utilizing separate dope solutions with different WSPs to each layer through the dual nozzle. Although the use of dual pore former had been previously studied by our group [26], this study utilized different pore formers in one dope solution phase. Our previous approach had the limited benefit of using environmentally friendly pore formers; to improve the process of employing distinct pore formers, the two WSPs were separately used in two dope solutions to fabricate the dual-layer HF membrane. PVP was used in the lumen side (1st layer), while the shell side (2nd layer) contained PEG. These WSPs possessed different rates of demixing in the polymer solution phase due to their intrinsic Hanssen solubility parameters and molecular weight. This could induce interactive diffusion of WSP in both directions (lumen side and shell side) when two-phase inversions occur. The fabricated HFs can form nanoscale pores via rapid demixing of PVP on the lumen side. Meanwhile, the PVP in the first dope solution (e.g., the 1st layer) could be drawn and diffused due to the swift diffusion in the direction of the shell side of PEG to form a structure with macrovoids and open pores. The fabricated HF membranes were expected to possess a high pure water flux and negligible leakage of Bovine serum albumin (BSA). Additionally, application of the studied HF membrane was confirmed by evaluating its performance on the urea clearance test.

## 2. Materials & Methods 

### 2.1. Dual-Layer Hollow-Fiber Membranes (DHF) Fabrication

#### 2.1.1. Preparation of Recipe and Configuration of Dual-Layer 

For the NIPS process, evenly dissolved poly(ethersulfone) (PES, Ultrason E 6020P, BASF, Ludwigshafen, Germany) in N-methyl-2-pyrrolidone with different additives (PEG or PVP) were prepared as dope solutions. The additives were polyethylene glycol (PEG, molecular weight = 400 Da, SAMCHUN PURE CHEMICAL Co., Ltd., Pyeongtaek, Korea) and polyvinylpyrrolidone k-30 (PVP, molecular weight = 40,000 Da, Samchun Chemical Cp., Ltd., Pyeongtaek, Korea). These additives were employed for the polymer mixture according to the ratios (wt.%) shown in Table 1. The blended solutions were stirred at 60 °C for one night. The bore solution used for forming a hollow structure in the center of the fiber was prepared by a simple stirring of NMP: deionized water (DI): glycerol (Extra Pure, 99.0%, DEAJUNG CHEMICALS & METALS Co., Ltd., Sihung, Korea) = 45:45:10 (wt.%). This solution was prepared at room temperature. The dope solution composition of fabricating dual-layer HF membrane (DHF) is presented in detail in Table 1. 

#### 2.1.2. DHF Spinning Process Preparation

The HF spinning process was conducted using a dual nozzle consisting of a co-extrusion of dope solution and bore solution. Before each fabrication process, all spinning conditions in Table 2 were set at the desired values. Since the flowrate of bore solution and dope solutions directly affect the scale of HF structure, the controlling factors were modulated in the range value displayed in Table 2. In detail, the faster spinning speed reduced the exposure of the extruded dope solution to the air gap before sinking in the coagulation tank. Meanwhile, the slower spinning speed increased the exposure of the extruded dope solution to the air gap. Changes in the pulled speed of the dope solution like these affected the surface morphology of HF. The spinning speed was also modulated by take-up winder speed in the range of 12–15 m/min.

#### 2.1.3. HFs Membrane Mini-Module Organization and Assembly

The mini-module consisted of perfluoroalkoxy tubing (ID: 3.9 mm, OD: 6 mm), T-fitting, and HFs. The packing density of HF modules was 10%, and the number of HFs was calculated using Equation (1). The effective membrane area of each HF module was obtained at 13.3, 12.3, and 13.2 cm^2^ for DHF-512, DHF-520, and DHF-528, respectively. A commercial membrane was also assembled with an effective area of 11.7 cm^2^ in the same way, namely a CHF. After the assembly of the module, the dead end of each side was filled with epoxy to harden, and then HF was fixed accordingly. Last, the modules were immersed in DI for stabilization.
(1)$$\begin{array}{c} \mathrm{Packing}\;\mathrm{density}\;(\%)=100\times n\times \frac{r^{2}}{R^{2}},\\ \mathrm{where}[\begin{array}{c} r\;\text{is the outer radius of hollow fiber}\\ R\;\text{is the inner radius of housing}\\ n\;\text{is the number of hollow fiber} \end{array} \end{array}$$

### 2.2. Membrane Characterization

#### 2.2.1. HF Membrane Morphology

The morphology of DHFs and a CHF membrane were characterized by using an ultra-high resolution field emission scanning electron microscope (UHR-FE-SEM, Verios 5 UC, Thermo Fisher Scientific Inc., Waltham, MA, USA). The specimens were prepared via the cryo-cutting method with liquid nitrogen (DAEDEOKGAS Co., Ltd., Incheon, Korea). The HFs were wet with ethyl alcohol (ethanol absolute, 99.9%, Ducksan, Seoul, Korea) and immersed in liquid nitrogen for rapid cooling. By bending frozen HFs, a hollow structure cross-section was obtained. Samples were vertically aligned beside the specimen stage and coated with Pt for 180 s to take the cross-sectional image. Pt coating for 60 s was applied to the lumen and shell sides of HFs by aligning the sample horizontally on the specimen stage.

#### 2.2.2. Porosity of HF Membranes

Porosity was measured to evaluate membrane structural properties. The dried HFs membranes were collected at the length of 10 cm, in bundles, and weighed on the mass balance. Next, HF membranes were immersed in isopropanol (IPA, 99.5%, OCI Co., Ltd., Seoul, Korea) for 24 h. Residual IPA was removed from the surface of HFs before the wet HFs were weighed on mass balance. To calculate the porosity of HF membranes, the dried and wet masses were substituted into Equation (2) below.
(2)$$\mathrm{Porosity},\;\epsilon \;\left(\%\right)=100\times \frac{\frac{m_{w}-m_{d}}{\rho_{w}}}{\frac{m_{w}-m_{d}}{\rho_{w}}+\frac{m_{d}}{\rho_{m}}},\;\mathrm{where}\;[\begin{array}{c} m_{w}\;\text{is mass of wet HFs with IPA}\;\left(g\right)\\ m_{d}\;\text{is mass of dried HFs}\;\left(g\right)\\ \rho_{w}\;\text{is density of IPA}\;\left(0.786\;g/{\mathrm{cm}}^{3}\right)\\ \rho_{d}\;\text{is density of HFs}\;\left(1.37\;g/{\mathrm{cm}}^{3}\right) \end{array}$$

#### 2.2.3. Pure Water Permeability (PWP) Test

The PWP test was conducted via a hydraulic filtration system, as shown in Figure 1a. Here, 4 L of DI was prepared as a feed solution, and a beaker was placed on the mass balance as a permeate collector. Mini-modules were pre-stabilized on the hydraulic filtration system with a flow rate in the range of 50–80 mL/min at 0.9 bar in 60 min. For the permeation portion of the mini-modules, one side was closed, and the other side was open for collecting permeate. After the system was ready for testing, the gear pump was set to circulate DI at a flowrate in the range of 50–80 mL/min at TMPs of 0.3, 0.6, and 0.9 bar. TMP value was determined using Equation (3). Each TMP condition was maintained for 40 min, and the permeate mass was recorded by the computer. The data were taken in the last 10 min in which the flux was stabilized. As expressed in Equation (4), pure water permeability (PWP) (mL/m^2^·h·mmHg) was determined by calculating the slope of a linear fitting between the water flux and TMP.
(3)$$\mathrm{Transmembrane}\;\mathrm{pressure}\;\left(\mathrm{bar}\right)=\frac{P_{\mathrm{Feed}\;\mathrm{in}}+P_{\mathrm{Feed}\;\mathrm{out}}}{2},\;\mathrm{where}\;[\begin{array}{c} P_{\mathrm{Feed}\;\mathrm{in}}\;\mathrm{is}\;\mathrm{inlet}\;\mathrm{flow}\;\mathrm{of}\;\mathrm{feed}\\ P_{\mathrm{Feed}\;\mathrm{out}}\;\mathrm{is}\;\mathrm{outlet}\;\mathrm{flow}\;\mathrm{of}\;\mathrm{feed} \end{array}$$
(4)$$\begin{array}{l} \mathrm{Pure}\;\mathrm{water}\;\mathrm{permeability}\;(L/m^{2}\cdot h\cdot \mathrm{bar})=\frac{\Delta J_{W}}{\Delta P},\\ \;\mathrm{Where}\;[\begin{array}{c} J_{W}\;\text{is Water flux}\;\left(L/m^{-2}\cdot h\right)\\ \Delta P\;\text{is transmembrane pressure}\;\left(\mathrm{bar}\right),\;\left(\mathrm{TMP}\right)\\ \mathrm{Where}\;\mathrm{linear}\;\mathrm{fitting}:\;J_{W}=\mathrm{PWP}*\Delta P+b \end{array} \end{array}$$

#### 2.2.4. Mean Pore Size Calculation 

The mean pore size could be estimated by following the Guerout–Elford–Ferry Equation (5) [28]; equation terms are HF membrane porosity, water flux at specific pressurized TMP, and HF membrane configuration. The radius of pores was calculated, and the pore size (diameter) of HF membranes was determined by multiplying by two. (5)$$\begin{array}{cc} \mathrm{Pore}\;\mathrm{size}\;(\mathrm{nm}) & =2r\\ & =\sqrt{\frac{\left(2.9-1.75\varepsilon \right)\times 8\eta \mathrm{lQ}}{\varepsilon \times A\times \Delta P}}\\ & \times 2,\mathrm{where}\;[\begin{array}{c} \varepsilon \;\text{is porosity of HFs membranes}\;\left(\%\right)\\ \eta \;\text{is viscosity of water}\;\left(8.9\times 10^{-4}\;\mathrm{Pa}\cdot s\right)\\ l\;\text{is the thickness of HFs membranes}\;\left(m\right)\\ Q\;\text{is the volume of permeate water per unit time}\;\left(m^{3}/s\right)\\ A\;\text{is the effective area of HFs membranes}\;\left(m^{2}\right)\\ \Delta P\;\text{is the operational TMP}\;\left(\mathrm{Pa}\right) \end{array} \end{array}$$

### 2.3. Membrane Performance Test

#### 2.3.1. Bovine Serum Albumin (BSA) Leakage

Bovine serum albumin (BSA, M.W. = 66,000 Da, Sigma Aldrich Inc., Burlington, MA, USA) leakage test was conducted via a hydraulic filtration system (Figure 1a). This process was similar to the PWP test; however, the feed solution was replaced with a BSA solution of 1000 mg/L. The desired concentration of feed solution was prepared using a stock solution of 2 g/L BSA. The BSA stock solution had been prepared 1 day before as it is time-consuming to prepare a well-dispersed BSA aqueous solution. After the mini-module was installed in the system, the well-dispersed BSA solution was put on the feed stirrer and circulated from ‘feed in’ to ‘feed out’ with a flowrate in the range of 50–80 mL/min at TMP 0.8 bar. Then, the permeate solution was collected by conical tube at 10 min intervals for 30 min. The BSA feed solution was gathered using the same procedure as a permeate solution. The BSA concentration of feed and permeate were determined by absorbance of ultraviolet–visible spectroscopy (UV–Vis, S-3100, SCINCO Co., Ltd., Seoul, Korea), measured at a wavelength of 280 nm. The absorbance was then substituted into the calibration curve to obtain the concentration of feed and permeate solution. The calibration curve was predetermined based on known concentrations (1000, 500, 250, 100, 50, 25, 12.5 mg/L) and corresponding absorbance at the wavelength of 280 nm. The concentration of BSA for feed and permeate solution was substituted into the equation below, Equation (6), to obtain the leakage ratio of BSA through the membrane.
(6)$$\mathrm{BSA}\;\mathrm{leakage}\;\left(\%\right)=100\times \frac{C_{P}}{C_{F}}\mathrm{Where}\;[\begin{array}{c} C_{P}\;\mathrm{is}\;\mathrm{the}\;\mathrm{concentration}\;\mathrm{of}\;\mathrm{permeate}\;\mathrm{solution}\;\;\left(\mathrm{mg}/L\right)\\ C_{F}\;\mathrm{is}\;\mathrm{the}\;\mathrm{concentration}\;\mathrm{of}\;\mathrm{feed}\;\mathrm{solution}\;\left(\mathrm{mg}/L\right) \end{array}$$

#### 2.3.2. Urea Clearance: An Application in Hemodialysis

The urea clearance test was performed under a closed circulation system consisting of two parts (blood circulation and dialysate circulation), as shown in Figure 1b. Both were comprised of a peristaltic pump and a pressure gauge for each line. The artificial blood was prepared with 25 g/L of BSA and 1 g/L of urea dissolved in 500 mL of phosphate-buffered saline (PBS, tablet, Sigma Aldrich Inc., USA) aqueous solution. The dialysate was prepared with 500 mL of PBS aqueous solution. Before the circulation of each part, 6 mL of artificial blood was gathered to determine the initial concentration of blood. The stabilized mini-module was installed by the connection of the blood part and dialysate part as shown in Figure 1c. Each flowrate of the peristaltic pump was set with the TMP being zero between blood and dialysate using Equation (7) below.
(7)$$\begin{array}{cc} \mathrm{TMP}\;(\mathrm{bar})= & 0\;\mathrm{bar}\\ & \;\;=\frac{P_{\mathrm{Blood}\;\mathrm{in}}+P_{\mathrm{Blood}\;\mathrm{out}}}{2}-\frac{P_{\mathrm{Dialysate}\;\mathrm{in}}+P_{\mathrm{Dialysate}\;\mathrm{out}}}{2} \end{array}$$

In one session, artificial blood and dialysate samples were collected every hour for four hours after each part had been circulated. Prior to urea analyses, both solutions were pre-treated by separating BSA using centrifugal filters (Amicon^®^ Ultra—4, Merck Millipore Ltd., Burlington, MA, USA) at 9000 R.P.M for 15 min. After that, the concentration of urea by time was analyzed by measuring absorbance at the wavelength of 203 nm via UV–Vis. These concentrations were then substituted into Equation (8) to calculate the total mass removal of urea from artificial blood.
(8)$$\begin{array}{l} \text{Total mass removal of Urea at the time},E_{\mathrm{Ut}}\;\left(\mathrm{mg}\right)\\ \;=M_{U}\\ \;-V_{\mathrm{Bt}}C_{\mathrm{But}},\;\mathrm{Where}\;[\begin{array}{c} M_{U}\;\text{is initial mass of urea dissolved in blood}\;\left(\mathrm{mg}\right)\\ V_{\mathrm{Bt}}\;\text{is the volume of blood at a certain time}\;\left(L\right)\\ C_{\mathrm{But}}\;\text{is urea concentration of blood at certain time}\;\left(\mathrm{mg}/L\right) \end{array} \end{array}$$

The clearance of uremic toxins was defined as a negative correlation between the amount of uremic toxins eliminated in blood and the average concentration of blood [29,30]. Urea clearance by mini-module was determined, and it was then converted using a normalized effective area ratio between the studied mini-module (12.3 cm^2^) and commercial module (1.8 m^2^). The converted urea clearance is expressed in Equation (9).
(9)$$\begin{array}{l} \text{The converted clearance of urea }\;C_{\mathrm{Ut}}\;\left(\mathrm{mL}/\min\right)\\ =\frac{A_{C}}{A_{M}}\times \frac{E_{\mathrm{Ut}}}{P_{\mathrm{Ut}}},\;\mathrm{Where}\;[\begin{array}{c} C_{\mathrm{Ut}}\;\text{is clearance of urea at certain time}\;\left(\mathrm{mL}/\min\right)\\ A_{C}\;\text{is an effective area of commercial module}\;\left(m^{2}\right)\\ A_{M}\;\text{is an effective area of mini module}\;\left(m^{2}\right)\\ E_{\mathrm{Ut}}\;\text{is amount of urea eliminate in blood at certain time}\;\left(\mathrm{mg}/\min\right)\\ P_{\mathrm{Ut}}\;\text{is mean blood concentration of urea at certain time}\;\left(\mathrm{mg}/\mathrm{mL}\right) \end{array} \end{array}$$

## 3. Results and Discussion

The use of dual nozzles with dual phases containing different WSP additives in separate dope solution is a potential method to fabricate a dual-layer hollow fibers for application to hemodialysis and others i.e., water/wastewater treatment. As shown in Figure 2, this fabricating approach facilitated the formation of HFs with open pores from the lumen side (10–20 nm size pores) to the shell side (sub-micro-size pores). We expect this HF structure to possess the potential for ever-widening open pores based on the different molecular weights and Hansen solubility parameters of WSPs used in each dope solution (Table 3). PVP in dope solution has a slightly lower diffusion rate than PEG due to its larger molecular weight (MW). However, compared to PEG, the Hansen solubility parameter of PVP is closer to the value of water. Due to the intrinsic property of PVP, rapid mixing could occur in the first layer, which favored the formation of nanoscale pores on the lumen side. The movement of PEG in the second dope solution was more free compared to PVP due to its smaller MW. However, the lower Hansen solubility of PEG may have facilitated the delayed demixing that occurred in the second layer, resulting in the formation of sub-micro-scale pores on the shell side. Additionally, PEG helped form finger-like pores that directly connected the shell side from the lumen side of DHFs. Consequently, open pores between the lumen side and shell side of DHFs with a well-modulated pore scale are expected not only to promote the diffusion of small-sized solutes but also to favor the retention of selective large-sized solutes [31,32,33].

To demonstrate the feasibility of the proposed approach to DHF fabrication, membrane properties (SEM, porosity, PWP) and membrane performance (BSA leakage test, urea clearance test) tests were conducted. A summary of the results is presented in Table 4. In the following sections, we discuss advanced fabrication via dual-pore formers and dual dope solution phases in detail.

### 3.1. Morphology of DHF and CHF Membrane via SEM

The well-fabricated DHF had the structure of long threads that were as thin as human hair. The shape and constitution of the fibers could hardly be observed by the naked eye. Therefore, SEM was adopted to check the hollow structure inside of the fibers as well as their morphology. The whole cross-sectional structure of DHFs and the CHF membrane are presented in Figure 3, which depicts well-rounded shapes with concentric circle hollow structures. All HFs showed a smooth circle line without bumpy or embossed surfaces on either the lumen side or shell side.

Although each DHF had a different dual dope phase recipe, the dimensions of all DHFs were well-controlled in error, bound by modulation of spinning conditions, which were investigated in our previous research [3]. The contrast difference is displayed in the middle of the membrane substrates of Figure 3a–c; we attributed this difference to the dual-phase extrusion of the dual nozzle. For HF dimensions, unlike the DHFs, the CHF was just barely visible to the naked eye. The dimensions (OD, wall thickness) of each HF are presented in Figure 4, corresponding with the content of WSP. The outer diameter (OD) of HFs were 286.9 ± 8.3, 298.1 ± 6.1, and 284.6 ± 4.8 μm for DHF-512, DHF-520, and DHF-528, respectively. The OD of CHF was smaller than the DHs membranes, being 244.3 ± 1.8 μm. The OD of DHF showed a slight hillock curve at PEG 20 wt.%, but it was still within error range. As the content of PVP was fixed at 5 wt.% and the spinning condition was unchanged, the size of the studied HF did not change significantly despite the increasing PEG content. A similar trend was also found in wall thickness (Figure 4b). This fact indicated that the membrane thickness could be formed on a uniform scale using the dual-phase extrusion process. Notably, all studied DHF membranes showed relatively similar wall thickness. Finally, the inner diameter (ID) of DHFs was larger than that of CHF; this is due to estimation from the subtraction of thickness from OD. 

As the results in Figure 4 show, the wall thickness of DHF-512, DHF-520, DHF-528, and CHF was 34.5 ± 2.6 μm, 33.2 ± 0.5 μm, 33.8 ± 0.7 μm, and 31.2 ± 4.8 μm, respectively. In comparison with CHF, the larger ID of the studied DHF membranes helped to minimize the fouling degree, favor higher flux, and promote a lower pressure drop inside the lumen side, i.e., decrease of turbulence flow and membrane resistance [34]. 

Furthermore, the estimation of the required number of fibers for modularization on a commercial scale was computed and summarized in Table 5. As shown in Table 5**,** approximately 27% of fibers were saved for DHF-520 compared to CHF, as the effective area of the module was set at the same value of 1.8 m^2^. This outcome highlighted that the higher ID had more economical benefits due to its module configuration and properties.

The finger-like pores, the shell, and the lumen side of DHF were observed via high magnification of the cross-section (see yellow lines in Figure 5). The selected high-magnification scanning of DHF-520 was compared with CHF. As shown in Figure 5a, for DHF, well-built finger-like pores were located between the lumen and shell side. Meanwhile, a discontinued finger-like structure was found in CHF (Figure 5b). The open pores with growing size from the lumen side to the shell side were noticeable for all membranes (Figure 5b,c for DHF-520, Figure 5e,f for CHF). The comparison of shell-side structure between DHF and CHF is presented in Figure 5b,e. The results indicate a conspicuous difference at the end of finger-like pores (see yellow line in Figure 5). The blue arrow, which has different thicknesses, refers to the route of solution flow. The thick blue arrows are representative of the large flow, and the thin ones stand for relatively low flow. 

As mentioned, the low flow gathered at the shell side of DHF-520, and the collected flow easily passed through the membrane due to a gradual increase in open pores. However, a reduced size of finger-like pores was found at the shell side for CHF; thus, the flow might be impeded in passing through the shell side. For the lumen side, a very thin and dense layer was developed due to the rapid demixing of PVP (Figure 5c). In contrast, as shown in Figure 5f, the loose and thick sponge-like structure was built on the lumen side of CHF. Such a thin and dense layer of DHF on the lumen side enabled an increase in the selectivity of large-sized solutes (protein) and promoted the diffusion of small-size solutes (urea). Figure 6 shows greater detail of how the pore construction of the shell side and lumen side were well-detected on the top view of DHF-520 and CHF.

The top view of the lumen and shell side of DHF-520 and CHF were displayed in Figure 6 to show its nano-scale pores on the lumen side and sub-micro-scale pores on the shell side. It was found that a uniform size of nano-scale pores was evenly distributed on the lumen side of DHF-520 (Figure 6a). Meanwhile, as indicated by Figure 6b, a non-uniform size of nano-scale pores was found on the lumen side of CHF. For DHF-520, the lumen side exhibited a pore size of 24–28 nm verticality and 56–97 nm horizontality, while CHF showed a lumen side with a pore size of 56–69 nm verticality and 70–152 nm horizontality. Such pore construction of both DHF-520 and CHF was similarly verified on the high magnification of the cross-section in Figure 5. Similarly, the pores connected to the finger-like pores near the shell side, which can be seen in the cross-section of Figure 5b,e, were observed from the top view of the shell side, as shown in Figure 6c,d. The shell side of DHF-520 possessed few micro-scaled holes and evenly distributed sub-micro pores. Although the shell side of CHF was also constructed with porous sub-micro pores, these pores were spread on the surface of the shell side. Moreover, there were some sections with dense phases on the surface of the CHF shell side, and no porous structure was detected. These top-view SEM analyses have tended to be consistent with the cross-sectional structural analysis of DHF-520 and CHF.

Through structural analysis, it was confirmed that the dual-phase co-extrusion of dope solution, applied with two different WSPs, enabled the formation of integral finger-like pores that connect the lumen side and shell side. The extended finger-like pores from the lumen to the shell side can act as microchannels for fluids and reduce transmembrane resistance [35].

### 3.2. Membrane Properties and BSA Retention: In Comparison with CHF 

The membrane properties were evaluated for specific applications. Such basic properties of DHFs, e.g., porosity, PWP, and pore size, were determined accordingly. The porosity evaluation results of DHFs and CHF processed with IPA were calculated by Equation (3) and are displayed on the graph, with the content of WSP, that is shown in Figure 7a. Remarkable porosity was exhibited for all DHFs due to the collaboration of dual WSP. This was a much higher figure compared to CHF of 85.6 ± 2.0%; the figures were 92.0 ± 1.0% for DHF-512, 93.8 ± 1.2% for DHF-520, and 92.2 ± 0.6% for DHF-528. The PWP results are shown in the following graph, Figure 7b, which demonstrates a positive tendency of permeability with increasing PEG content. Even though there was no significant difference in porosity or membrane thickness among DHFs, a positive trend for PWP emerged as well. This was due to the increase of hydrophilic and open pores, which were induced from the lumen to the shell side as per WSP content. PWP property values were record as 188.0 ± 14.5 L/m^2^·h·bar for DHF-512, 213.5 ± 4.0 L/m^2^·h·bar for DHF-520, 296.2 ± 5.7 L/m^2^·h·bar for DHF-528, and 248.0 ± 19.9 L/m^2^·h·bar for CHF. The studied DHF membranes showed a higher PWP compared to HF membranes obtained in previous works, such as single-layer 29.32–98.32 L/m^2^·h·bar in [36]; single-layer 198 L/m^2^·h·bar in [37]; and dual-layer 3.95 L/m^2^·h·bar in [38].

Moreover, these outcomes directly affected the mean pore size of DHFs, as shown in Figure 7c. Notably, the increase in PEG instigated the diffusion of PEG from second phase to the bore solution. The mean pore size of DHF-512, DHF-520, DHF-528, and CHF was 26.8, 27.4, 33.2, and 31.7 nm, respectively. The nano-scale pores of the lumen side could be modulated by controlling the PEG content. The BSA leakage results, which are shown in Figure 7d, were influenced by changing mean pore size. For DHFs, mean pore size increased as the content of PEG increased in the second layer. The scale of pore size being affected by the content of second layer PEG was expected due to excess PEG, resulting in a lower Hansen solubility parameter than the PVP. This led to diffusion to the lumen side and decreased the rate of demixing. The measured values of BSA leakage were 0.06 ± 0.02% for DHF-512, 0.17 ± 0.08% for DHF-520, 1.74 ± 0.03% for DHF-528, and 2.16 ± 0.02% for CHF. DHF-528 showed a lower leakage of BSA than CHF. We attribute this to the smaller pore size of DHFs compared to CHF. As presented in Figure 6a,b, DHFs possessed more uniformity of pore formation compared to CHF. The superiority of the spinning process with co-extrusion of a dual dope solution including dual WSP was proved. 

Notably, in de Fierro, 2015 [39], a paper from the blood-treatment membrane field, membrane properties could be improved with the large pore size and BSA leakage of CHF compared to DHF-520, exhibited by reduced mean pore size and decreased BSA leakage after blood contact due to the clogging of the blood corpuscles. In the case of the above-mentioned phenomenon, our membrane, which had a similar pore size but higher PWP and lower BSA leakage, DHF-528 might be expected to show improved properties over CHF after blood contact due to its uniform pore size.

### 3.3. Urea Clearance by the Studied Membranes

Given their remarkable properties—smooth and well-formed hollow structures with lumen-to-shell finger-like pores, large porosity, high PWP, uniform pores, and low BSA leakage—DHFs were considered for application as blood treatment membranes or protein chromatography membranes. Among these, urea clearance was tested using DHFs for confirmation of the superiority of DHFs.

Urea clearance was computed from the amount of total urea removal, including DHF-520, which has shown balanced properties in membrane property evaluation. The removal of urea and leakage of BSA were measured every hour; these results are displayed in Figure 8. As shown in the graph, BSA leakage value remained low with high removal of urea. After one hour, the urea removal volume was remarkable, at around 27,452 mg/m^2^. Even though there was a stagnation in the removal of urea after 2 h, removal began to increase again 3 h later. After 4 h of operation (one session), the total removal of urea reached 34,626 mg/m^2^. To compare with CHF, the total volume of urea removal and average concentration of synthetic human plasma were substituted into Equation (9) to calculate the clearance of urea in one session. The areal factor was standardized by the effective area ratio between the mini-module (12.3 cm^2^) and commercial module (1.8 m^2^). The urea clearance of DHF-520 and CHF was 257.6 mL/min and 281.9 mL/min, respectively; the results are presented in Table 4.

## 4. Conclusions

This work successfully fabricated an HF membrane with a dual layer containing different WSPs. The fabricated DHF membranes exhibted a uniform nano pore size (26.8–33.2 nm) on the lumen side and and opened pore size (sub-micro) on the shell side. By utilizing different WSPs (PVP in first layer and PEG in second layer), the properties of the dual layer could be independently controlled, leading to an outstanding structure of finger-like, controlled pore size on both sides. The open pore structure was verified by cross-sectional SEM images, while the superior performance of DHFs was confirmed by the outcomes of high PWP (188–296 L/m^2^·h·bar). 

Additionally, the mean pore size of the lumen side could be controlled by increasing the PEG content (from 12–28%). The outstanding properties of the fabricated DHFs were also clearly demonstrated in the outcomes for BSA leakage and urea clearance. Overall, compared to CHF, our fabricated DHFs showed a minimized BSA leakage (DHF-520: ~0.20%) while possessing a comparable urea clearance (DHF-520: 257.6 mL/min, CHF: 281.9 mL/min).

## Figures and Tables

**Figure 1 membranes-12-01140-f001:**
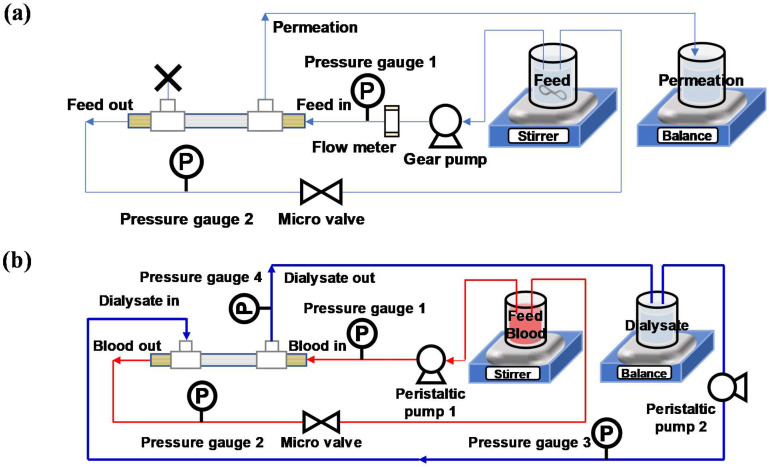
The schematic diagrams of (**a**) the hydraulic filtration system; (**b**) the closed-circulation urea clearance test system.

**Figure 2 membranes-12-01140-f002:**
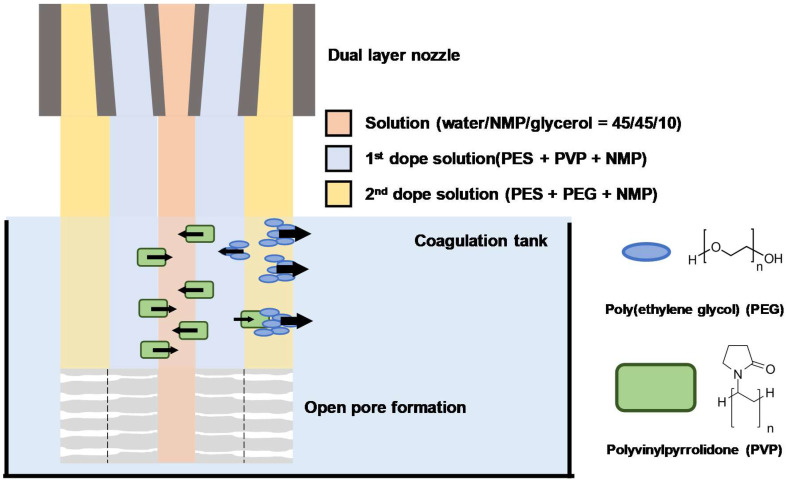
The schematic diagram of dual-layer spinning of dual-pore former that includes dual-phase dope solution via dual-layer nozzle.

**Figure 3 membranes-12-01140-f003:**
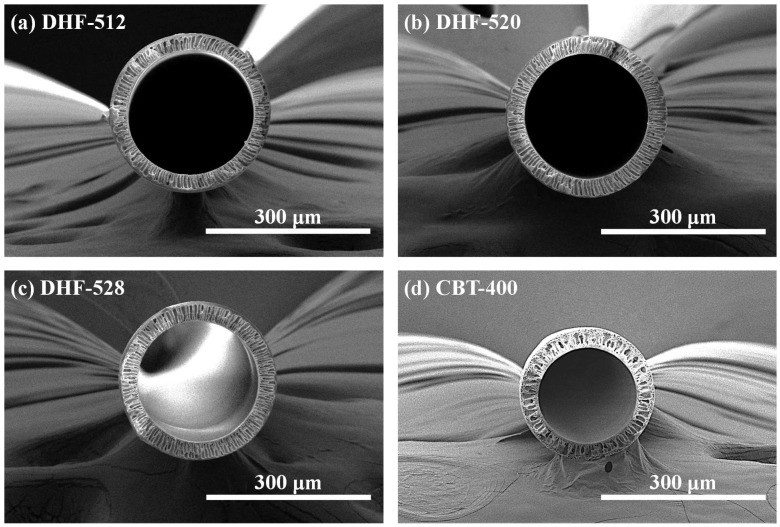
Cross-sectional images of DHFs and CHF, (**a**). 12 wt.% of PEG content HF referred to DHF-512, (**b**). 20 wt.% of PEG content HF referred to DHF-520, (**c**). 28 wt.% of PEG content HF referred to DHF-528 (**c**), and CHF (**d**).

**Figure 4 membranes-12-01140-f004:**
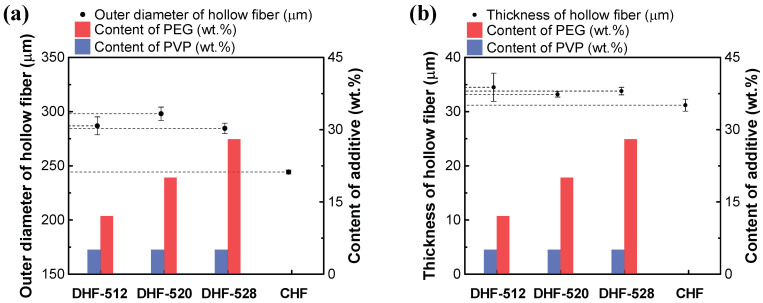
Comparison of membrane structure based on PEG contents: (**a**) OD of DHFs and CHF, (**b**) membrane thickness of DHFs and CHF. The membrane ID could be deduced from the difference between OD and thickness.

**Figure 5 membranes-12-01140-f005:**
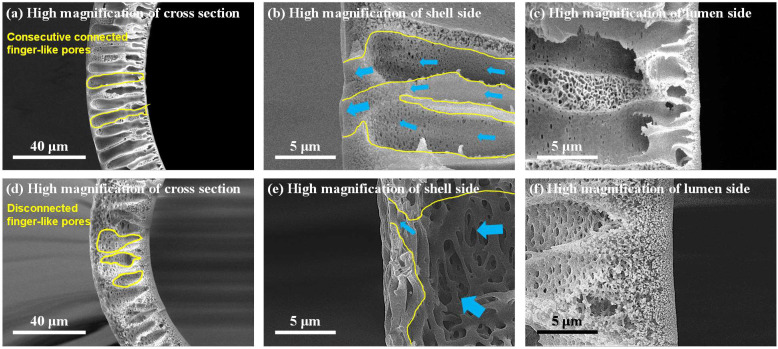
Confirmation of the formation of finger-like open pores connected with nanopores on the lumen side. (**a**). High magnification of DHF-520 cross-section, (**b**). High magnification of DHF-520 shell side, (**c**). High magnification of DHF-520 lumen side, (**d**). High magnification of CHF cross-section, (**e**). High magnification of CHF shell side, (**f**). High magnification of the CHF lumen side.

**Figure 6 membranes-12-01140-f006:**
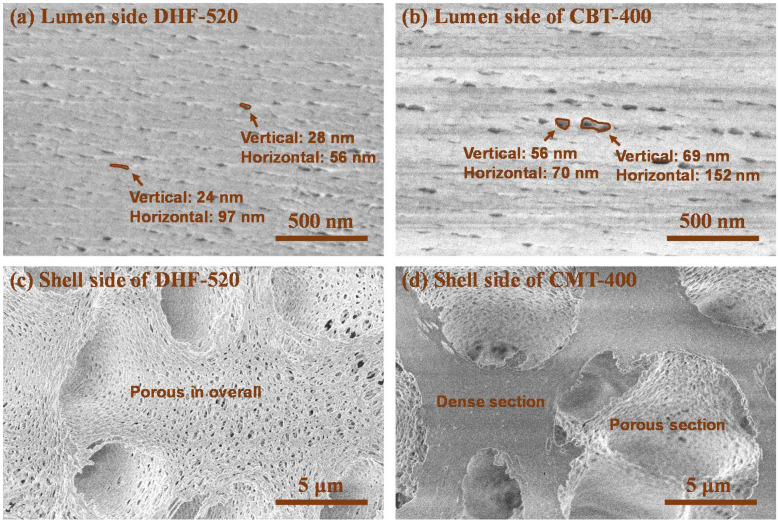
Surface morphology of lumen side and shell side, (**a**). High-magnification top-view SEM image of DHF-520 lumen side, (**b**). High-magnification top-view SEM image of CHF lumen side, (**c**). Shell-side top-view SEM image of DHF-520. Shell-side top-view SEM image of CHF, (**d**).

**Figure 7 membranes-12-01140-f007:**
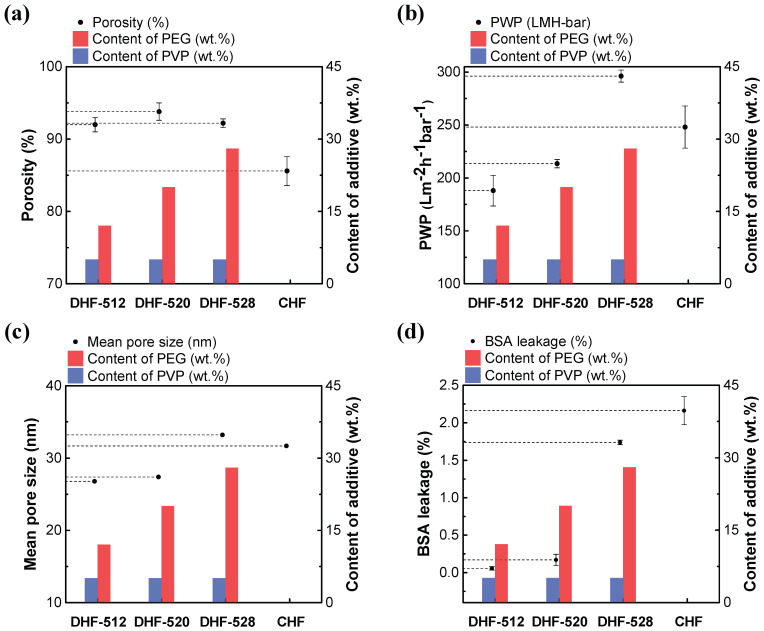
Comparison of membrane properties based on the contents of additives. (**a**) The porosity of DHFs and CHF. (**b**) Pore size of DHFs and CHF. (**c**). Pure water permeability (PWP) of DHFs and CHF. (**d**) BSA leakage of DHFs and CHF.

**Figure 8 membranes-12-01140-f008:**
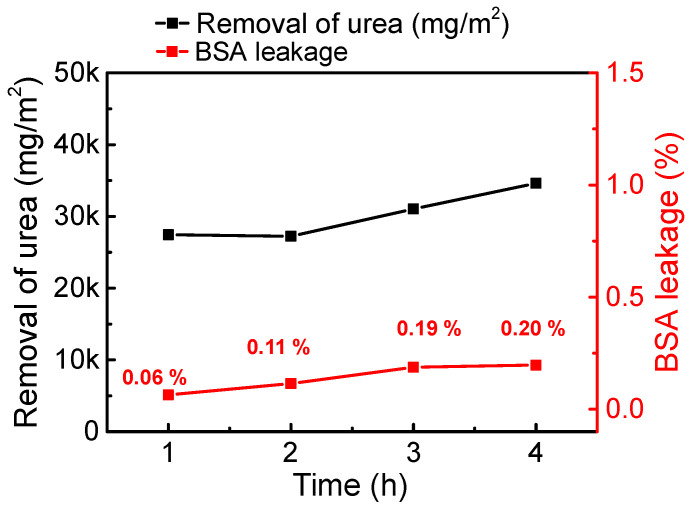
Removal of urea from synthetic human plasma by DHF-520 membrane during one session.

**Table 1 membranes-12-01140-t001:** Dope solution composition.

Classification		Water-Soluble Pore Former (WSP, wt.%)		N-Methyl-2-pyrrolidone (NMP, wt.%)	Poly-Ether Sulfone (PES, wt.%)
DHF-512	1st dope	PVP	5	81	14
	2nd dope	PEG	12	74	
DHF-520	1st dope	PVP	5	81	
	2nd dope	PEG	20	66	
DHF-528	1st dope	PVP	5	81	
	2nd dope	PEG	28	58	

**Table 2 membranes-12-01140-t002:** HF membrane fabrication spinning condition.

Factors	Values
Bore solution supply (mL/min)	0.6–0.8
1st dope solution supply (R.P.M.)	1–1.5
2nd dope solution supply (R.P.M.)	1–1.5
Air gap (cm)	20–30
Take-up winder speed (m/min)	12–15
Coagulation tank temperature (°C)	25
Nozzle temperature (°C)	40–55
Dope tank temperature (°C)	40–55

**Table 3 membranes-12-01140-t003:** Intrinsic properties of components in dope solution and non-solvent.

Classification	Molecular Weight (M.W.)	Hansen Solubility Parameter ($\delta_{t}$, MPa^1/2^)
Water	18.02 g/moL	47.80 [31]
N-methyl-2-pyrrolidone (NMP)	99.13 g/moL	22.90 [32]
Polyvinylpyrrolidone k-30 (PVP)	40,000 Dalton	21.57 [33]
Polyethylene glycol (PEG)	400 Dalton	18.90 [31]

**Table 4 membranes-12-01140-t004:** Properties of DHF and CHF membranes.

Classification	WSP (wt.%)		Outer Diameter (μm)	Thickness (μm)	Mean Pore Size (nm)	Porosity (%)	Pure Water Permeability (L/m^2^·h·bar)	BSA Leakage (%)	Urea Clearance (mL/min)
	PVP	PEG							
DHF-512	5	12	286.9 ± 8.3	34.5 ± 2.6	26.8	92.0 ± 1.0	188.0 ± 14.5	0.06 ± 0.02	None
DHF-520	5	20	298.1 ± 6.1	33.2 ± 0.5	27.4	93.8 ± 1.2	213.5 ± 4.0	0.17 ± 0.08	257.6 *
DHF-528	5	28	284.6 ± 4.8	33.8 ± 0.7	33.2	92.2 ± 0.6	296.2 ± 5.7	1.74 ± 0.03	None
CHF	No information		244.3 ± 1.8	31.2 ± 1.1	31.7	85.6 ± 2.0	248.0 ± 19.9	2.16 ± 0.19	281.9 [30] **

* The urea clearance test result was obtained from 1 session for 4 h in a closed circulation system, and this value was converted to correspond to the effective surface area of the commercial module of 1.8 m^2^. ** The urea clearance test results in a commercial module were processed in a clinical test, 1 session for 4 h.

**Table 5 membranes-12-01140-t005:** Estimation of the required number of fibers to reach the assumption of a module with effective length of 0.25 m and membrane area of 1.8 m^2^.

Classification	DHF-520	CHF
Inner diameter (ID) (m)	2.316 × 10^−4^	2.443 × 10^−4^
Effective area of HF (m^2^)	1.819 × 10^−4^	1.429 × 10^−4^
Required number of fibers	≈9896 fibers	≈12,599 fibers

## Data Availability

Not applicable.

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
