# Peer review of "Open Pore Ultrafiltration Hollow Fiber Membrane Fabrication Method via Dual Pore Former with Dual Dope Solution Phase"

_membranes, 2022, doi:10.3390/membranes12111140_

Round 1

Reviewer 1 Report

This work fabricated dual-layer hollow fiber membranes for ultrafiltration experiments. Some interesting and significant works were carried out in this work. However, this manuscript suffers from some shortcomings. And some improvements should be made before the paper meets the requirement for publication.

The authors should pay more attention to the following points.

(1)  More work is needed to improve the level of English throughout the entire manuscript.

(2)  Does the molecular weight of PEG affect the membrane structure and performance?

(3)  Error bars should be added in Figure 8.

(4)  How about the service life of the membrane?

(5)  Some comparison between the results y in this work and those reported in literature should be added.

Author Response

Response to Reviewer 1 Comments

Title: Open pore ultrafiltration hollow fiber membrane fabrication method visa dual pore former with dual dope solution phase

We are very grateful to the Editor and the reviewers for the valuable suggestions and constructive comments, which have helped us improve the quality of the manuscript. Our detailed responses to the reviewers’ comments follow below and revisions in the manuscript text are highlighted in yellow.

Reviewer #1: This work fabricated dual-layer hollow fiber membranes for ultrafiltration experiments. Some interesting and significant works were carried out in this work. However, this manuscript suffers from some shortcomings. And some improvements should be made before the paper meets the requirement for publication. The authors should pay more attention to the following points.

Point 1: More work is needed to improve the level of English throughout the entire manuscript.

Response 1: We have improved English for the whole manuscript (MS). The revisions/corrections were highlighted in the revised MS.

Point 2: Does the molecular weight of PEG affect the membrane structure and performance?

Response 2: We appreciate this question. It has been reported that the molecular weight of PEG can affect the membrane structure and performance of UF membranes (JMS, 2006, p. 920-927; JMS, 2008, p. 209-221; Desalination, 2011, p.51-58). These works just focused on fabricating a single layer of HF membrane. Their study indicated that pure water permeability and porosity of membranes increase with an increase in molecular weight of PEG; however, adding a high MW of PEG can result in larger pore size of the skin layer; thus, decreasing a BSA rejection. MW of PEG plays a substantial role in precipitation. With an increase in the molecular weight of PEG, more PEG molecules are permanently trapped in the membrane because of their lower mobility after phase separation in the coagulation bath. As a result, the polymer lean phase becomes less in quantity and larger in vacancy which results in the formation of larger pores in the skin layer. For our study, although PEG is used in the outer layer, PEG can also diffuse into the inner side and cause affecting on the properties of the inner surface (pore size). A low MW of PEG (400 Da) is selected for study to obtain a higher BSA rejection (less BSA loss), which is favorable for hemodialysis. However, we also acknowledge the reviewers' question, this fact helps us think about further investigating the effect of different PEG on the properties of an outer layer in a dual-layer HF membrane.  

Point 3: Error bars should be added in Figure 8.

Response 3: Thanks for your helpful suggestion. We just conducted a duplicate experiment. Therefore, the value shown in Figure is average. Typically, for statistical data, an error bar should be added as a triplicate experiment is conducted.

Point 4: How about the service life of the membrane?

Response 4: We acknowledge this comment. Testing the cycle stability of the hollow fiber membranes is just for the application to water treatment/water desalination, which can indicate the service life of the membrane. However, we aim to fabricate the membrane for the hemodialysis application. Typically, a dialyzer (hemodialysis membrane) is used only once per patient during treatment (for 4-6 h operation). This operation is also clarified by previous works studying hemodialysis membranes in clinical tests. Therefore, based on the experiment of urea clearance, our membrane is totally to meet this requirement.

Point 5: Some comparison between the results in this work and those reported in literature should be added.

Response 5: Thanks for your nice suggestion. Comparisons with the previous works have been added and discussed in MS in terms of PWP

   ⇒ This modification can be found on page 21 (lines 404-407).

Reviewer 2 Report

The paper is well-written, with minor corrections needed. For example, 

-line 140- space needed between assembled_with

-line 139- no need for dots after numbers

-line 135- should be assembly instead of ssembly

The author should go through the entire paper and correct all of the spelling mistake or errors 

The authors should play with the brightness of sem images. They could do it on PowerPoint and then paste them into the manuscript. 

Author Response

Response to Reviewer 2 Comments

Title: Open pore ultrafiltation hollow fiber membrane fabrication method visa dual pore former with dual dope solution phase

We are very grateful to the Editor and the reviewers for the valuable suggestions and constructive comments, which have helped us improve the quality of the manuscript. Our detailed responses to the reviewers’ comments follow below and revisions in the manuscript text are highlighted in yellow.

Reviewer #2:  The paper is well-written, with minor corrections needed. For example,

Point 1: Line 140- space needed between assembled with

Response 1: We have corrected it accordingly on the manuscript (MS).

Point 2: Line 139- no need for dots after numbers

Response 2: We have checked. Removing dots was corrected accordingly.

Point 3: line 135- should be assembly instead of ssembly

Response 3: We have corrected this mistake. Additionally, the spelling mistake or errors was revised for the whole MS. The revision was highlighted in MS.

Point 4: The author should go through the entire paper and correct all of the spelling mistake or errors

Response 4: Thanks for going through this in detail MS, we have corrected all of the spelling mistakes or errors. The reivison was highlighted in MS.

Point 5: The authors should play with the brightness of SEM images. They could do it on PowerPoint and then paste them into the manuscript.

Response 5: Thanks for this suggestion. The brightness of SEM images was adjusted accordingly.

Reviewer 3 Report

Reviewer #: Title: Open pore ultrafiltration hollow fiber membrane fabrication method via dual pore former with dual dope solution phase (membranes-2002417)

This manuscript is an article about the Dual-layer Hollow Fiber Membranes Fabrication control pores and internal structure of hollow fiber membrane by fabricating a dual-layer using a dual nozzle. I would like to recommend the minor revision, please check the following comments:

1)     #The two WSPs have different rates of demixing due to differences in Hanssen solubility parameters, and different diffusion rates in the polymer solution phase due to differences in molecular weight. This behavior of WSP induces interactive diffusion of WSP in both directions (lumen side and shell side) when two-phase inversions occur.# please further explain the influence of Hanssen solubility and molecular weight on the two-phase inversion, by discussing the PVP and PEG in this work.

2)     Why the authors choose Solution (water/NMP/glycerol = 45/45/10) as pore-forming solution?

3)     As discussed in Figure 7, Effect of rate of extrusion on the pores size should be discussed, which is important for industrialization.

Author Response

Response to Reviewer 3 Comments

Title: Open pore ultrafiltation hollow fiber membrane fabrication method visa dual pore former with dual dope solution phase

We are very grateful to the Editor and the reviewers for the valuable suggestions and constructive comments, which have helped us improve the quality of the manuscript. Our detailed responses to the reviewers’ comments follow below and revisions in the manuscript text are highlighted in yellow.

Reviewer #3: This manuscript is an article about the Dual-layer Hollow Fiber Membranes Fabrication control pores and internal structure of hollow fiber membrane by fabricating a dual-layer using a dual nozzle. I would like to recommend the minor revision, please check the following comments:

Point 1: The two WSPs have different rates of demixing due to differences in Hanssen solubility parameters, and different diffusion rates in the polymer solution phase due to differences in molecular weight. This behavior of WSP induces interactive diffusion of WSP in both directions (lumen side and shell side) when two-phase inversions occur.# please further explain the influence of Hanssen solubility and molecular weight on the two-phase inversion, by discussing the PVP and PEG in this work.

Response 1: Thanks for the helpful comments. This fact is also raised by reviewer 1. Actually, it was hypothesized that Hanssen solubility and molecular weight of both WSP on the two-phase inversion could cause different demixing rates. Our following experiments (Section 3.1) supported this hypothesis through the outcomes of pore size and morphology of Hollow fibers (HFs) (SEM photos in Fig. 5, 6).

It is noted that the effect of molecular weight of PEG on the inner structure and performance of UF membrane had been reported by a past work (JMS, 2006, p. 920-927). Their findings indicated that the increase of PEG molecular weight could result in larger pore size of the skin layer due to their lower mobility after phase separation. PEG's lower mobility increases the polymer lean phase rather than the polymer-rich phase, and the lean phase changes to a void, thereby enlarging the pores. Furthermore, NMP has a closer Hansen solubility parameter to water than PEG. Therefore, with lower solubility of PEG to non-solvent (water), it implies that the PEG does not easily diffuse to non-solvent in the polymer phase compared to NMP. This fact results in promoting the polymer lean phase as PEG composition increases; thereby enlarging the pore size. In our study, PEG was used for fabricating the outer layer of HFs, and the outcomes of this study (large pore size and open pore at the outer surface supported these statements). Meanwhile, PVP is the first dope (inner layer) has a slightly lower diffusion rate than PEG due to its larger molecular weight (MW). Due to the intrinsic property of PVP, a rapid demixing could occur in the 1st layer, which favored the formation of nanoscale pores on the lumen side.

  • Phenomenon combined with the experimental results via control composition of 2nd phase PEG with a fixed composition of 1st phase of PVP were presented on pages 12-19.

Point 2: Why the authors choose Solution (water/NMP/glycerol = 45/45/10) as pore-forming solution?

Response 2: We highly appreciate this question. Actually, our fabricated membrane is a target for scaling up mass production. For a small-scale, glycerin is normally used for post-treatment of HFs, which helps HFs more flexible (avoiding shrinkage). However, for the mass production scale, immerging HFs into glycerin solution seems to be not convenient for continuous mass production.

Bore solution is a crucial factor affecting to morphology and pore size of HF. We selected a ratio of NMP: Water= 1:1 to control a proper pore size, which was studied by our recent work (JMS, 2022, 663,121065) Therefore, in order to minimize the influence of pore size and morphology by the bore solution when transferring to the mass production system, the bore condition of the mass production system was used.

In considering mass production, glycerin was also added to the bore solution (10 wt%) to help the membrane avoid shrinkage during drying without immerging process. In this process, the glycerol is remained on the whole surface of the inner membrane structure and prohibits the pore size to be shrunk. The shrunk pore size leads to a decrease in the pure water permeability and this result can be found in the preliminary data below (DHF-520 fabrication with and without treating glycerol). As the result, the porosity also decreased as DHF-520 was fabricated without the glycerol immerging process.

Sample

Porosity (%)

Pure water permeability (L/m2∙h∙bar)

DHF-520 immerged in glycerol

93.8 ± 1.2

213.5 ± 4.0

DHF-520 without glycerol immerging process

91.3 ± 0.5

64.5 ± 14.7

Point 3: As discussed in Figure 7, Effect of rate of extrusion on the pores size should be discussed, which is important for industrialization.

Response 3: We totally agree with the reviewer that the flow rate of extrusion has a considerable effect on the formation of the pore size of HFs. This fact was investigated by our recent works ((JMS, 2022, 663,121065). Our previous outcomes also highlighted that spinning conditions including dope solution supply rate, bore solution supply rate, winding speed, dope solution temperature, air gap, coagulation temperature, and dope solution composition affect the membrane pore structure. Notably, the rate of extrusion directly influences the outer diameter (OD), inner diameter (ID), and wall thickness of HFs and these properties had a significant impact on pore size distribution. A low flow rate of 1.0-1.5 mL/min designated in this study is to fabricate an HF membrane with thin wall thickness, and a smaller OD (increasing packing density), which favors hemodialysis application.
